# An Acidic Heteropolysaccharide Isolated from *Pueraria lobata* and Its Bioactivities

**DOI:** 10.3390/ijms24076247

**Published:** 2023-03-26

**Authors:** Shifan Zhao, Hualei Xue, Yijiong Tao, Kai Chen, Xiao Li, Mi Wang

**Affiliations:** Key Laboratory of Veterinary Chemical Drugs and Pharmaceutics, Ministry of Agriculture, Shanghai Veterinary Research Institute, Chinese Academy of Agricultural Sciences, Shanghai 200241, China

**Keywords:** *Pueraria lobata*, polysaccharide, structure, bioactivity

## Abstract

A novel water-soluble acidic heteropolysaccharide, called PPL-1, was purified from *Pueraria lobata*. PPL-1 had an average molecular weight of 35 Kad, and it was composed of glucose, arabinose, galactose and galacturonic acid (6.3:0.8:0.8:2.1). In accordance with methylation and nuclear magnetic resonance analyses, PPL-1 primarily consisted of (1→2)-linked α-Araf, (1→4)-linked α-Glcp, (1→)-linked β-Glcp, (1→6)-linked α-Glcp, (1→3,6)-linked α-Galp, (1→)-linked β-GalpA and (1→4)-linked α-GalpA. In terms of bioactivities, PPL-1 exhibited remarkable scavenging ability towards DPPH (1,1-Diphenyl-2-picrylhydrazyl) radicals and moderate activity by enhancing the proliferation rate of RAW 264.7 cells by approximately 30% along with the secretion of NO. This work demonstrates that PPL-1 can be a potential source of immunoenhancers and antioxidants.

## 1. Introduction

*Pueraria lobata*, the dried root of kudzu or ‘Ge-Geng’ in Chinese, is a well-known traditional Chinese medicine for treating common cold, headache and hypertension and for alleviating alcohol intoxication. It has been extensively used as food in southern China for thousands of years. Studies have found that the effective constituents of *Pueraria lobata* are isoflavones such as puerarin, which was isolated in the late 1950s and used in treating diabetes and cardiovascular and cerebrovascular diseases [1,2]. By contrast, research on another abundant active ingredient of *Pueraria lobata*, namely, polysaccharide, remains limited.

Most studies on plant polysaccharides have focused on their activities and structural characteristics. Plant polysaccharides exhibit strong bioactivities, including antioxidant, antitumour, immunomodulatory, hypoglycaemic and hypolipidemic activities [3,4,5]; they also have complicated structures. For example, the 36 types of polysaccharides from *Angelica sinensis* have different monosaccharide compositions, molecular weights (Mws), positions of glycosidic linkages and bioactivities [6]. Similar examples are ginseng and *Lycium barbarum* [7]. The aforementioned findings may be attributed to the structural characteristics of plant polysaccharides being affected by raw materials and extraction and purification methods [8]. Different polysaccharide components from the same herb have been reported and investigated, improving the understanding of the relationship between structure and biological activity and facilitating the development and application of herbal medicine in modern medicine [6,9,10,11].

Studies have shown that polysaccharides from *Pueraria lobata* exhibit various bioactivities, such as good antioxidant and immunomodulatory activities and remarkableα-amylase inhibitory activity [12,13,14,15]. Moreover, we have previously reported that polysaccharides from *Pueraria lobata* can significantly increase the richness and diversity of the gut microbial community in mice [16]. The structural characterisations of polysaccharides from *Pueraria lobata* have been investigated from three aspects: Mw, monosaccharide composition and glycosidic linkage. Five of the polysaccharides extracted from *Pueraria* root were glucans, whilst the remaining polysaccharides were determined to contain galactose, mannose, fructose, xylose and arabinose through monosaccharide composition analysis; however, detailed studies, such as those on glycosidic linkage, remain lacking [15,17,18,19].

In the current study, an acidic heteropolysaccharide, called PPL-1, was purified from *Pueraria lobata*. The structural characterisation and in vitro effects of PPL-1 on lipopolysaccharide (LPS)-induced inflammation have not been reported previously. This work aimed to enrich the knowledge regarding the structural characterisation and antioxidant and immunomodulatory activities of polysaccharides from *Pueraria lobata* to promote their potential applications in the food and pharmaceutical industries.

## 2. Results and Discussion

### 2.1. Results

#### 2.1.1. Isolation and Colourimetric Analysis of PPL-1

PPL-1 was collected in accordance with the elution curve of eluent, which was detected using the phenol–sulphuric acid method on a Sephadex G-100 column, as shown in Figure 1A. The yield was 28%. The total carbohydrates and uronic acid of PPL-1 were measured at 64% and 21%, respectively. Moreover, the UV spectrum (Figure 1B) indicated that PPL-1hadno protein content.

#### 2.1.2. Average Mw and FTIR Spectrum of PPL-1

The average Mw of PPL-1 was 35 kDa as determined via HPLC-GPC (High Performance Liquid Chromatography-Gel Permeation Chromatography). IR spectroscopy is a common method for identifying organic functional groups. As shown in Figure 1C, the strong and broad absorption peak at 3385 cm^−1^ and the relatively weaker peak at 2934 cm^−1^ were respectively attributed to the O–H and C–H stretching vibrations, which were considered the characteristic absorption bands of polysaccharides. The absorption peaks at 1740 cm^−1^ and 1620 cm^−1^ were the C=O stretching vibrations in the acetyl groups and the COO− deprotonated carboxylic group, respectively [20]. Combined with the strong and broad absorption peak at 3385 cm^−1^, a conclusion could be drawn that uronic acid was present. The absorption peaks in the region of 600–1500 cm^−1^(typically called the fingerprint region) were used to reflect the IR spectral differences of the same compound with varying structures. In the current study, the absorption band at 1420 cm^−1^ was the C–H bending vibration. The absorption peaks at 1149, 1080 and 1023 cm^−1^ indicated that C–O bonds and pyranose rings were present in PPL-1 [21]. The peak at 765 cm^−1^ was the characteristic absorption of Se=O in accordance with the published literature [19]. Compared with the IR spectra of the other polysaccharides isolated from *Pueraria lobata*, the absorption bands of PPL-1 in the fingerprint region (1500–600 cm^−1^) was approximate [17,18,19].

#### 2.1.3. Monosaccharide Composition Analysis of PPL-1

The PMP (1-phenyl-3-methyl-5-pyrazolone) pre-column derivatisation method can generate quantitative reactions with carbohydrates under mild conditions and facilitate the separation of monosaccharides via reversed-phase liquid chromatography by employing C18 as the stationary phase; combined with the sensitivity and specificity of UPLC–MS (Ultra Performance Liquid Chromatography-Mass Spectrum) analysis, the monosaccharide analysis results could be obtained [22]. The results of monosaccharide composition analysis are presented in Figure 2 and Table 1. PPL-1 was largely composed of *glucose*, *arabinose*, *galactose* and *galacturonic acid*, with a molar ratio of 6.3:0.8:0.8:2.1. The amounts of xylose and rhamnose were minimal (no quantitative data).

#### 2.1.4. Glycosidic Linkages of PPL-1

The glycosidic linkages of PPL-1 are presented in detail in Table 2. A conclusion could be drawn that glucose, galactose, arabinose and galacturonic acid residues were obtained. The signal of 1,4,5-tri-O-acetyl-2,3,6-tri--O-methyl glucitol was the highest peak. It indicated that the presence of (1→4)-linked Glcp units was a major part of PPL-1 (50.1%). Then, 1,4,5-tri-O-acetyl-2,3,6--tri-O-methyl galactitol and 1,5-di-O-acetyl-2,3,4,6-tetra-O-methyl galactitol were assigned to (1→4)-linked GalpA (13.1%) and t-linked GalpA (7.7%), respectively. Lastly,1,5-di-O-acetyl-2,3,4,6-tetra-O-methylglucitol,1,3,5,6-tetra-O-acetyl-2,4-di-O-methyl galactitol and 1,5,6-tri-O-acetyl-2,3,4-tri-O-methyl glucitol were assigned to t-linked Glcp (7.3%), 1→3,6-linked Galp (11.2%), (1→2)-linked Araf (5.4%) and (1→6)-linked Glcp (5.1%), respectively. In addition, the numbers of terminal, branch and linear residues were 4, 3 and 18, respectively. Thus, the degree of branching (DB) value was 28%.

#### 2.1.5. NMR (Nuclear Magnetic Resonance) Analysis of PPL-1

^1^HNMR and ^13^CNMR are important tools for interpreting the structure of PPL-1, such as identifying the configuration of the glycosidic bond (α or β) and the sugar ring (pyranose or furanose). In accordance with the chemical shift of the end matrix sub-signal, the α configuration is larger than δ_H_ 5 ppm, whilst the β configuration is smaller than δ_H_5 ppm. From the ^1^HNMR spectra (Figure 3A), seven anomeric hydrogen signals appeared at δ_H_ 4.5–6 ppm. The chemical shifts δwere5.10 ppm(A/H1), 5.80 ppm (B/H1), 4.50 ppm (C/H1), 4.93 ppm (D/H1), 5.23 ppm (E/H1), 4.65 ppm (F/H1) and 5.24 ppm (G/H1). In the ^13^CNMR spectrum, anomeric carbon signals appeared at δ 90–110 ppm. From the correlation between ectopic hydrogen and carbon signals in the heteronuclear single quantum correlation (HSQC) spectrum (Figure 3D), the chemical shifts of the two can be matched one-to-one: 5.10/107.41 ppm(A/H1-C1),5.80/106.84 ppm (B/H1-C1), 4.50/102.96 ppm (C/H1-C1), 4.93/99.86 ppm (D/H1-C1), 5.23/95.88 ppm (E/H1-C1), 4.65/95.75 ppm (F/H1-C1) and 5.24/91.96 ppm (G/H1-C1). Amongst them, residues C (4.50/102.96 ppm) and F(4.65/95.75 ppm)had β configuration, whilst the remaining residues had α configuration. In our studies, the a forementioned signals were assigned toH1/C1 of (1→2)-linked α-Araf, (1→4)-linked α-Glcp, t-linked β-Glcp, (1→6)-linked α-Glcp, (1→3,6)-linked α-Galp, (1→4)-linked β-GalpA and t-linked α-GalpA (Table 3), in accordance with previously reported data [23,24,25,26]. On the basis of the correlation of ^1^H–^1^H in the ^1^H–^1^H COSY spectrum (Figure 3C), the 2–6 hydrogen related to the aforementioned end-group hydrogen were assigned, and the corresponding 2–6 carbon were assigned via the HSQC spectrum. All carbon and hydrogen signals are provided in the Table 3.

In the heteronuclear multiple-bond correlation (HMBC) spectrum (Figure 3E), the cross peak signals around δ_H_ 5.10/δ_C_ 70.14 ppm (A H1/B C3), δ_H_ 3.55/δ_C_ 91.96 ppm (A H2/G C1), δ_H_ 580/δ_C_ 70.14 ppm (B H1/B C4), δ_H_ 4.97/δ_C_ 106.84 ppm (E H3/B C1), δ_H_ 3.25/δ_C_ 95.88 ppm (B H4/E C1), δ_H_ 4.67/δ_C_ 70.14 ppm (F H1/B C4), δ_H_ 3.25/δ_C_ 102.96 ppm (B H4/C C1), δ_H_ 3.25/δ_C_ 99.86 ppm (B H4/D C1), δ_H_ 5.24/δ_C_ 72.25 ppm (G H1/D C6)and δ_H_ 4.97/δ_C_ 95.75 ppm (E H3/F C1) proved the correlation between them. A conclusion could be drawn that GalpA(1→2)-Araf(1→4)-Glcp(1→4)-Glcp(1→, GalpA(1→6)-Glcp(1→4)-Glcp(1→ and Glcp(1→4)-Glcp(1→ branches of carbohydrate chain exist in PPL-1. Simultaneously, they were connected to the same carbohydrate chain by →4)-Glcp(1→6)-Galp(1,3→. The methylation analysis results were consistent with those of the preceding results. Combined with the methylation analysis results, we predicted the structures of this carbohydrate chain and the repeating units of PPL-1 as shown in Figure 3G.

#### 2.1.6. Antioxidant and Immunomodulatory Activity of PPL-1

From the results of the DPPH assay (Figure 4), PPL-1 demonstrated excellent DPPH radical scavenging activity with a concentration-dependent relationship. In addition, we also evaluated the immunomodulatory activity towards RAW 264.7 cells. Micrographs of RAW 264.7 murine macrophages treated without LPS (Lipopolysaccharide) (A) or with LPS(B)are shown in Figure 5. The results showed that the cells treated with LPS increased in size and showed obvious pseudopodia. Firstly, the effects of PPL-1 at various concentrations (0.02, 0.10, 0.39, 1.56, 6.25, 25, 100 and 400 μg/mL) on cell survival was expressed with the relative proliferation rate (Figure 6A). PPL-1 could promote cell proliferation at concentrations of 0.02–25 μg/mL, reaching the highest of about 30% at a concentration of 1.56 μg/mL, compared with the control group (cells grown in an environment containing only medium). Secondly, the addition of PPL-1 could significantly promote the secretion of NO at concentrations of 1.25–10 μg/mL compared with that in the control group (Figure 6B). Lastly, PPL-1 exhibited no differences with the LPS group at a concentration of 0.01–0.1 μg/mL and even higher at a concentration of 1–10 μg/mL (Figure 6C,D).

### 2.2. Discussion

PPL-1 exhibited significant differences in Mw, monosaccharide compositions, glycosidic linkages and biological functions compared with previous polysaccharides isolated from *Pueraria lobata*, which might be due to the differences in extraction and purification.

To obtain the structural characterisation of polysaccharides, common methods used included average Mw determination, monosaccharide composition analysis, methylation analysis, IR spectral analysis and NMR spectroscopy. In the previous studies, the major monosaccharide constituent of *Pueraria lobata* polysaccharides was glucose, and different glycosidic linkages were detected via methylation analysis [12,14,17,18], including →4)-D-Glcp-(1→, →6)-D-Glcp-(1→, →3)-D-Glcp-(1→ and →4,6)-D-Glcp-(1→ or →3,6)-D-Glcp-(1→. Moreover, xylose, galactose, mannose, fructose, arabinose and uronic acid were detected in *Pueraria lobata* polysaccharides without further methylation analysis and NMR spectroscopy analysis [15,19]. In our study, amylase was used to degrade the starch in *Pueraria lobata* polysaccharides and then purified to obtain acidic heteropolysaccharide PPL-1 that consisted of glucose, arabinose, galactose and galacturonic acid at the ratio of6.3:0.8:0.8:2.1. The predominant glycosidic residues came from Glc (62%) and GalA (20%), which were close to the results of the monosaccharide composition analysis. In particular, galacturonic acid is the second most abundant monosaccharide component in PPL-1, and it may affect the activity of PPL-1.

Apart from monosaccharide composition and glycosidic linkages, the bioactivities of polysaccharides are considerably affected by their average Mws, and this phenomenon might be due to the different digestion and fermentation efficiencies of polysaccharides with different Mws. Dou et al. compared the digestive property and bioactivity of blackberry polysaccharides with different Mws; they found that all polysaccharides exhibited similar effects on gut microbiota, but they were easily utilised by bacteria with lower Mw [20]. Researchers have evaluated the repair effect of *Gracilaria lemaneiformis* sulphated polysaccharides with different Mws on damaged renal epithelial cells and found that 49.6 kDa exhibited the best repair effect on oxalate-induced damaged HK-2 cells [27]. To date, the average Mw range of the isolated polysaccharides from *Pueraria lobata* was from 2.6 kDa to 385 kDa. In accordance with previous studies on bioactivities, the isolated polysaccharides from *Pueraria lobate* demonstrated remarkable immunomodulatory activities, such as enhancing the activity of the maturation of murine dendritic cells through TLR4 signalling [13], increasing pinocytic and phagocytic capacities and promoting the secretion of NO, IL-6 and TNF-α with membrane receptors of GR, SR and TLR4 [12], good antioxidant activity by hydroxy, DPPH radical scavenging capacity and outstanding hypoglycemic activity via α-amylase inhibitory activities [26].

In our previous studies, we evaluated the antioxidant activity of polysaccharides with different volumes of ethanol treatment and gel filtration with different carriers. The optimal choice we obtained was used to extractPPL-1. Its structural characterisation and in vitro antioxidant and immunomodulatory activities were investigated. The results were further compared with those of the previous literature. The DPPH radical is widely used to evaluate the free radical scavenging activity of antioxidants. In this study, PPL-1 exhibited better DPPH radical scavenging activity than the polysaccharides from *Pueraria lobata* reported by Wang et al. [15]. This result could be attributed to the lower Mw and higher content of uronic acid in PPL-1. Uronic acid and Mw are related factors with regard to DPPH radical scavenging activity: the former may be beneficial for the antioxidant activity of polysaccharides through the activation of the hydrogen atom of the anomeric carbon; the latter presented a negative correlation with DPPH radical scavenging activity in many articles [28,29,30]. 

The strengthening of the immune system helps the body resist the threats of exogenous pathogens, and polysaccharides may activate effector cells, such as macrophages, to secrete NO and cytokines that modulate the immune system [31]. The results showed that PPL-1 not only enhanced the cell proliferation rate by about 30% at 1.56 μg/mL but also significantly increased NO production in RAW 264.7 cells in a dose-dependent manner (1.25–10 μg/mL) compared with the control group (Figure 6A). As the positive control group, the LPS treatment group can induce inflammatory damage of cells (Figure 5B). By contrast, PPL-1 did not over activate RAW 264.7 cells (Figure 6B). Nevertheless, NO concentration was increased under the intervention of low-concentration PPL-1 (0.01–0.1 μg/mL) but presented no significant difference compared with the LPS group. At high concentrations (1–10 μg/mL), it increased significantly. Combined with the PPL-1-alone treatment assay, we speculated that the high concentrations (1–10 μg/mL) of PPL-1not only had no protection and treatment activities but also had a significant strengthening effect on the LPS promotion of NO secretion in RAW 264.7 cells. Thus, the protection and treatment activities of PPL-1 may be further investigated after modification, such as acetylation or sulphuration [32].

### 2.3. Conclusions

In summary, we extracted an acidic heteropolysaccharide from *Pueraria lobata* (PPL-1). PPL-1 was composed of glucose (63%), arabinose (8%), galactose (8%) and galacturonic acid (21%). The major linkage types of PPL-1 were proven to be (1→2)-linked α-Araf,(1→4)-linked α-Glcp, (1→)-linked β-Glcp, (1→6)-linked α-Glcp, (1→)-linked α-Galp, (1→3,6)-linked α-GalpA and (1→4)-linked β-GalpA. PPL-1 can be a potential source of immunoenhancers and antioxidants.

## 3. Materials and Methods

### 3.1. Reagents

The monosaccharide standards (D-Glucose, D-arabinose, D-xylose, D-galactose and L-rhamnose) were purchased from Dr. Ehrenstorfer GmbH (Germany). D-galacturonic acid and N-cyclohexyl-N-[2-(4-methyl-1-oxa-4-azoniacyclohex-4-yl)ethyl]methanediimine,4-methylbenzenesulfonic acid were bought from Sigma-Aldrich (Shanghai, China). Trifluoroacetic acid (TFA) was procured from Aladdin Industrial Corporation (Shanghai, China). Methanol, acetonitrile and ammonium acetate, with high-performance liquid chromatography (HPLC) grade, were acquired from Fisher Scientific (Shanghai, China). 1-Phenyl-3-methyl-5-pyrazolone (PMP), 2,2-diphenyl-1-picrylhydrazyl (DPPH), 3-(4,5-dimethylthiazol-2-yl)-2,5-diphenyltetrazolium bromide (MTT), dimethyl sulphoxide (DMSO) and other analysis materials were purchased from Sinopharm Chemical Rea-gent Co., Ltd. (Shanghai, China). Dulbecco’s Modified Eagle’s Medium (DMEM), foetal bovine serum (FBS) and phosphate-buffered solution (PBS) were procured from Thermo Fisher Scientific (Shanghai, China). The nitric oxide (NO) detection kit was bought from Solarbio Science & Technology Co., Ltd. (Shanghai, China). Sephadex G-100 was obtained from Yuanye Co., Ltd. (Shanghai, China). The macrophage RAW 264.7 cell line was acquired from the Cell Bank of the Shanghai Institutes for Biological Sciences, Chinese Academy of Sciences, Shanghai.

### 3.2. Preparation of the Polysaccharides

The dried roots of *Pueraria lobata* were obtained from Shanghai Lei yun shang Pharmaceutical Co., Ltd. (Shanghai, China). Crude polysaccharides were extracted in accordance with a previously described method [16,33]. The starch in the crude polysaccharide was decomposed by a-amylase. Crude polysaccharides (30 mg) were dissolved in 1 mL of water, and then 0.5 mL of the solution was loaded onto a pre-equilibrated Sephadex G-100 column (1.6 cm × 60 cm). The column was washed with water at a flow rate of 1.5 mL/min. The eluent was detected in accordance with the phenol–sulphuric acid method [34]. Then, it was collected and concentrated to obtain PPL-1.

### 3.3. Colourimetric Analyses

The total carbohydrates and uronic acid of PPL-1 were respectively measured using the phenol–sulphuric acid and *m*-hydroxybiphenyl methods [35]. Ultraviolet-visible (UV2800, Shanghai, China) spectrometry was performed to detect protein contents.

### 3.4. Determination of Average Mw

HPLC-GPC (waters alliance 2695, Waters 2414 Refractive Index Detector, USA) was conducted to determine the average Mw of PPL-1. Commercially available dextrans with different molecular masses (1, 5, 10, 50, 150, 200 and 400 kDa) were used to establish the calibration curve in accordance with [36]. The sample was dissolved in water and then filtered through a 0.45 μm mesh before injection.

### 3.5. Infrared (IR) Spectrum Analysis

The functional groups of PPL-1 were detected using the KBr disk method combined with Fourier-transform IR spectroscopy (FTIR, Nicolet iS10, USA, Thermo Fisher Scientific, Waltham, MA, USA). A mixture of PPL-1 (2–3 mg) and KBr powder was pressed into a pellet and then scanned within the wavenumber range of 500–4000 cm^−1^.

### 3.6. Monosaccharide Composition

PPL-1 (10 mg) was hydrolysed in 20 mL of 2 mol/L TFA for 6 h at 100 °C. The hydrolysate was dried in nitrogen atmosphere and then washed three times with methanol to remove excess TFA. Then, the dried samples were redissolved in water to a concentration of 1 mg/mL. The PPL-1 hydrolysate was diluted to 100 µg/mL before derivatisation. Glucose, arabinose, xylose, galactose, rhamnose and galacturonic acid were used as monosaccharide standards. These standards were configured at a concentration of 1 mg/mL and then serially diluted to concentrations ranging from 0.15 ng/mL to 30 µg/mL. Then, 100 µL of PPL-1 hydrolysate and monosaccharide standards were mixed with 200 µL of ammonia solution in water and 200 µL of 0.5 M PMP solution in methanol. The mixtures were allowed to react at 70 °C for 30 min and then dried in nitrogen atmosphere. The dried samples were reconstituted in 500 µL of water and 500 µL of chloroform. Then, the dried samples were shocked, and the chloroform layer was discarded. The extraction process was repeated three times to remove excess PMP. The aqueous layer from each sample was stored at −20 °C before detection. The resulting solution was analysed through UPLC–QqQ MS (Waters, Westminster, SC, USA) by using a C18 column (2.1 mm × 100 mm, 1.7 μm). The chromatography and mass spectrometry conditions were optimised in accordance with Amicucci et al. [37]. In particular, the mobile phase A for UPLC parameters was 25 mM ammonium acetate (adjusted to pH 8.2 with NH_4_OH)–acetonitrile (95:5, *v*/*v*), whilst the mobile phase B was 25 mM ammonium acetate (adjusted to pH 8.2 with NH_4_OH)–acetonitrile (5:95, *v*/*v*). Flow state was set at 0.3 mL/min, and column temperature was set at 35 °C. Gradient elution was performed as follows: 0–7 min, 12–15% B; 7–12 min, 15% B; 12–13 min, 99% B; and 13–15 min, 12% B.

### 3.7. Glycosidic Linkages

PPL-1 (10 mg) was dissolved in 1 mL of deionised water. Then, 1 mL of carbodiimide was added (100 mg/mL) and allowed to react for 2 h. Subsequently, 1 mL of imidazole (2 M) was added to the samples, and the mixture was divided into two parts. Thereafter, 1 mL of NaBH_4_ (30 mg/mL) was added to one part, whilst 1 mL of NaBD_4_ (30 mg/mL) was added to the other part. The solutions were allowed to stand for 3 h. Then, 100 μL of glacial acetic acid was added to terminate the reaction, and the samples were freeze-dried after being dialysed for 48 h. Lastly, the samples were redissolved in DMSO and then methylated with methyl iodide for 1 h. Subsequently, they were hydrolysed with TFA and acetylated with acetic anhydride to obtain partially methylated alditol acetates (PMAAs), which were detected via gas chromatography−mass spectrometry (GC−MS,7890A-5977B, Agilent, Santa Clara, CA, USA). The glycosidic linkages of PPL-1 were identified by comparing with the literature and the database of the Complex Carbohydrate Research Center (CCRC, University of Georgia, USA). The molar ratio of each linkage residue was calculated on the basis of peak areas.

### 3.8. Nuclear Magnetic Resonance (NMR) Analysis

PPL-1 (30 mg) was dissolved in 0.5 mL of D_2_O. NMR spectroscopy of ^1^H, ^13^C, DEPT-135, COSY, HSQC and HMBC were analysed using a Bruker 700 MHz NMR apparatus (Avance Neo 700 MHz) (Instrumental Analysis Center of Shanghai Jiao Tong University, Shanghai, China).

### 3.9. DPPH Radical Scavenging Activity of PPL-1

DPPH radical scavenging activity was adopted to evaluate the antioxidant activity of PPL-1. Firstly, PPL-1 was dissolved in deionised water at various concentrations (0.5–2.0 mg/mL), and then DPPH was dissolved in 95% ethanol at a concentration of 1 mM. An equal volume of solutions was mixed and allowed to react at room temperature (25 °C) for 30 min in the dark and measured at 517 nm. DPPH radical scavenging activity was calculated using the following equation:scavenging activity (%) = [1 − (A1 − A0)/A2] × 100%
where A1, A0 and A2 are the absorbance values of the sample group (DPPH and sample solutions), blank group (ethanol and sample solutions) and control group (DPPH and deionised water), respectively. Furthermore, Vc was used as a positive antioxidant reference.

### 3.10. Culture of RAW 264.7 Cells

The RAW 264.7 cell line was grown in DMEM with 10% FBS and cultured at 37 °C and 5% CO_2_. The cells grew well and were distributed as islands. The cells were normally round before the experiments (Figure 5A).

#### 3.10.1. Effects of PPL-1 on RAW 264.7 Cell Viability via MTT Assay

A concentration of 5 × 10^4^ cells/mL was added to a 96-well plate at 100 μL/well. After 2 h, PPL-1 with different concentrations (0–400 μg/mL) was used to replace the original medium in the wells for 24 h. Then, 10 μL of MTT (5 mg/mL) was added to the wells for 4 h. The liquid in the wells was absorbed and discarded. Thereafter, 150 μL of DMSO was added to dissolve formazans, and the plate was measured at 490 nm by Microplate Reader (Multiskan MK3, USA, Thermo Fisher Scientific).

#### 3.10.2. Effects of PPL-1 on the Production of NO in RAW 264.7 Cells

RAW 264.7 cells were adjusted to a concentration of 2 × 10^5^ cells/mL, seeded on 24-well plates for 2 h and then replaced with different concentrations of PPL-1 (0–10 μg/mL) or LPS (1.2 μg/mL) for 24 h. The liquid in the well was used to detect NO concentration in different treatment groups. In addition, we evaluated the protection and therapeutic effects of PPL-1 on RAW 264.7 cells that were stimulated with LPS. In the first experiment, cells were loaded onto 24-well plates in the same manner and stimulated with LPS (1.2 μg/mL) for 5 h. The medium was removed and added with different concentrations of PPL-1 (0–10 μg/mL) for 24 h. In another experiment, the process was similar, but the cells were treated with PPL-1 (0–10 μg/mL) for 24 h and then replaced with LPS (1.2 μg/mL) for 5 h.

### 3.11. Statistical Analysis

All the experiments were conducted in triplicate. Data were analysed using one-way ANOVA, followed by Tukey’s test for multiple comparisons (*p* < 0.05).

## Figures and Tables

**Figure 1 ijms-24-06247-f001:**
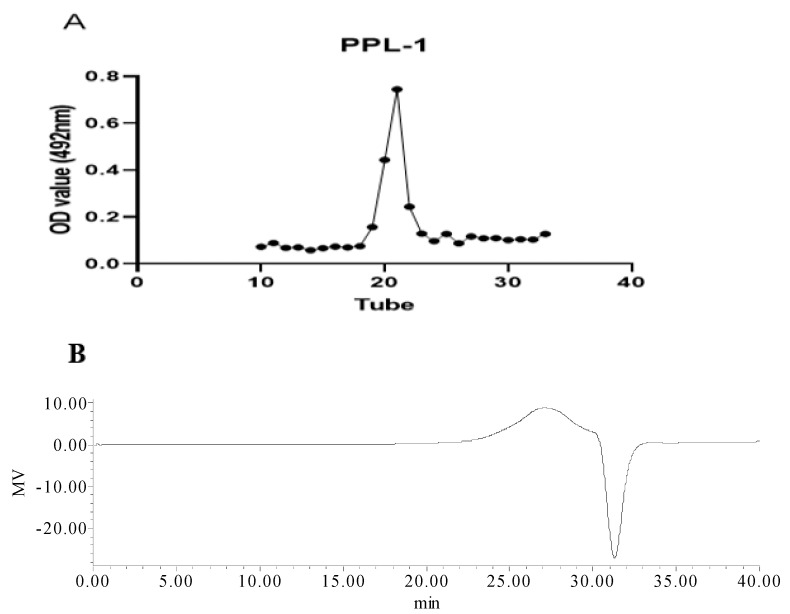
Elution curve (**A**) of PPL-1 on a SephadexG-100 column. UV spectrum (**B**) of PPL-1andFTIR spectrum (**C**).

**Figure 2 ijms-24-06247-f002:**
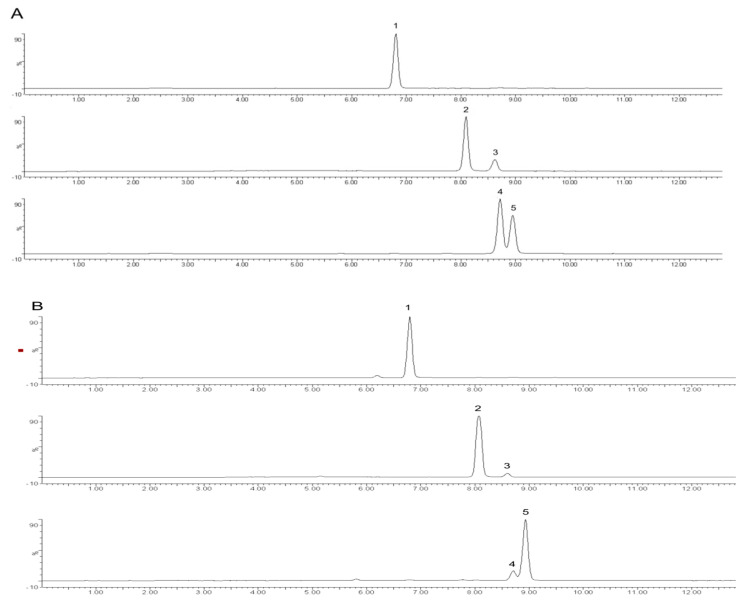
Multichannel single-ion monitoring chromatograms of the LC–MS of the PMP derivatives of mixed standard monosaccharides (**A**) and PPL-1 (**B**). Peaks: 1, galacturonic acid; 2, glucose; 3, galactose; 4, xylose; 5, arabinose.

**Figure 3 ijms-24-06247-f003:**
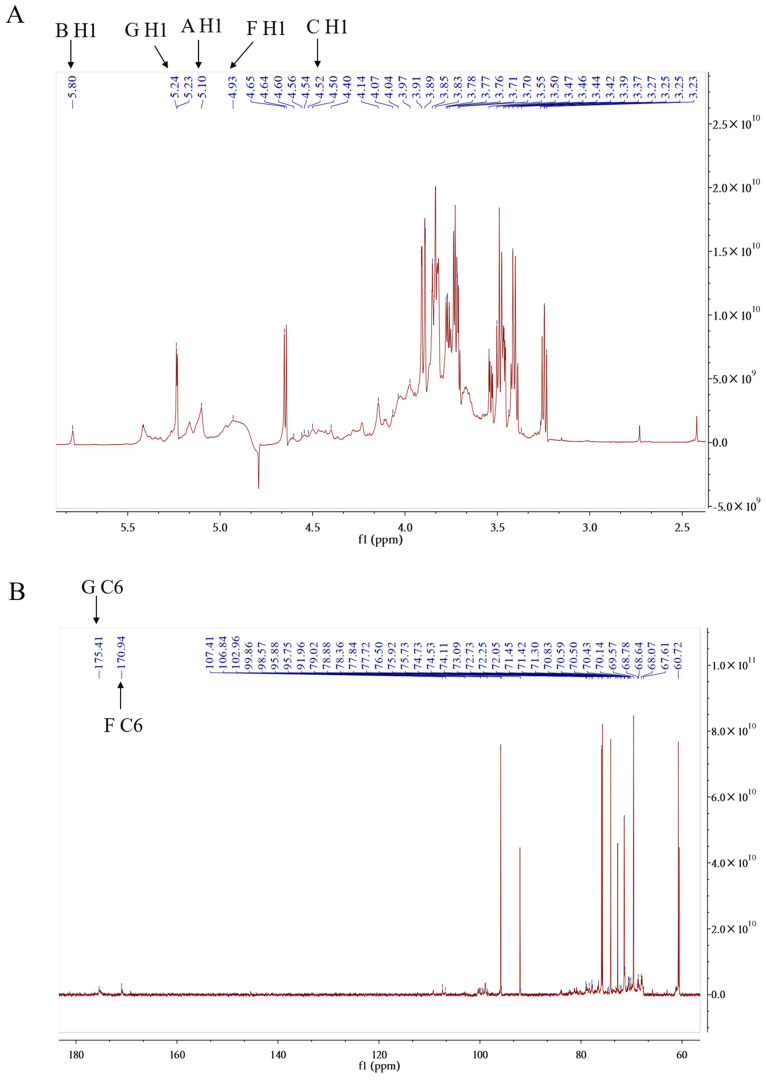
NMR spectra of PPL-1: ^1^HNMR (**A**),^13^CNMR (**B**),^1^H–^1^HCOSY (**C**), HSQC (**D**), HMBC (**E**) and DEPT-135° (**F**). Predicted structure of the repeating units of PPL-1 (**G**). (**F**) In the DEPT-135° spectrum, the first-order carbon atoms (CH3) and third-order carbon atoms (CH) are the upward peaks, the second-order carbon atoms (CH2) are the downward peaks and there is no peak of fourth-order carbon atoms (**C**).

**Figure 4 ijms-24-06247-f004:**
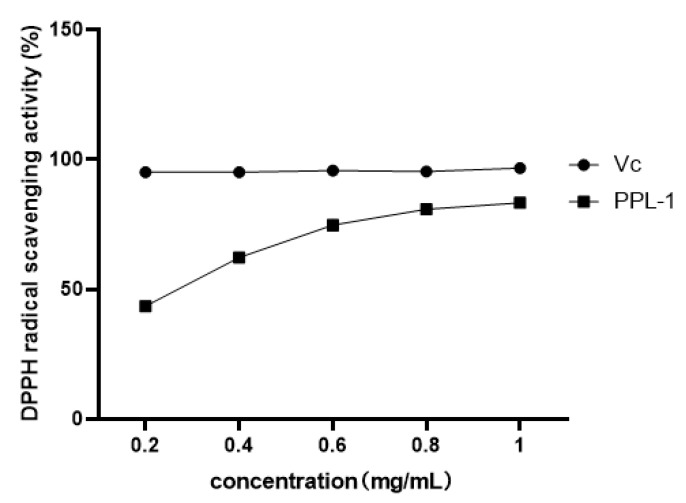
DPPH radical scavenging activity of PPL-1. Vcis the positive control group.

**Figure 5 ijms-24-06247-f005:**
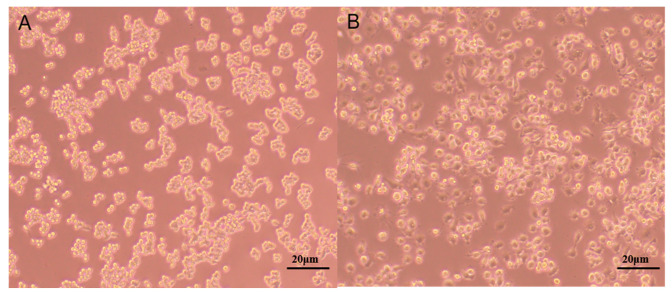
Micrographs of RAW 264.7 murine macrophages treated without LPS (**A**) or with LPS (**B**).

**Figure 6 ijms-24-06247-f006:**
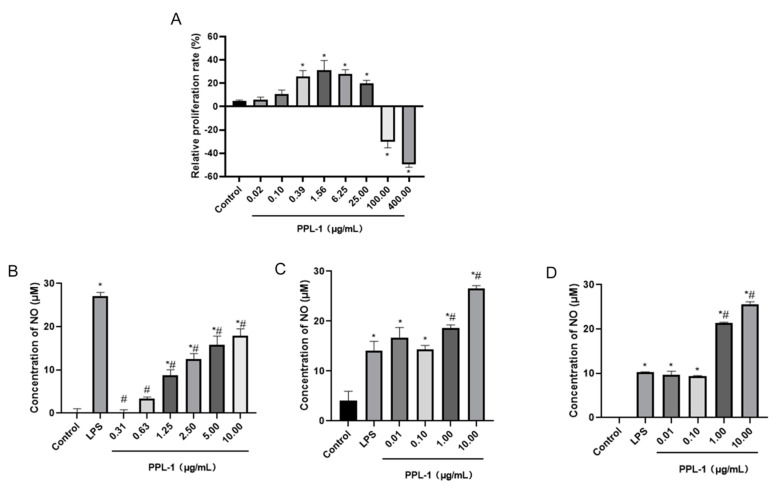
Effects of PPL-1 on the proliferation of RAW264.7 macrophages (**A**). (**B**–**D**) NO production secretion of RAW264.7 macrophages: (**B**) cells were treated with PPL-1 (0.01–10 μg/mL) and LPS (1.2 μg/mL) as the positive control group; (**C**) cells were treated with PPL-1 (0.01–10 μg/mL) after LPS (1.2 μg/mL) induction; (**D**) cells were induced by LPS (1.2 μg/mL) after being treated with PPL-1 (0.01–10 μg/mL). * *p* < 0.05 vs. the control group; ^#^
*p* < 0.05 vs. the LPS group (positive control group).

**Table 1 ijms-24-06247-t001:** Monosaccharide Composition and Molar Ratio of PPL-1.

Monosaccharide	Arabinose	Glucose	Galactose	Galacturonic Acid
Retention time (min)	8.9	8.1	8.6	6.8
Molar ratio (%)	8	63	8	21

**Table 2 ijms-24-06247-t002:** Deduced Glycosidic Linkage Type and Composition of PPL-1. Determined via Methylation and GC–MS Analyses.

Monosaccharide	Type of Linkage	Sugar Derivatives	Retention Time	Molar (%)
Arabinose	(1→2)-Araf	1,2,4-tri-O-acetyl-3,5-tri-O-methyl arabinitol	9.3	5.42
Glucose	(1→4)-Glcp	1,4,5-tri-O-acetyl-2,3,6-tri-O-methyl glucitol	14.4	50.11
	(1→)-Glcp	1,5-tri-O-acetyl-2,3,4,6-tetra-O-methyl glucitol	9.46	7.35
	(1→6)-Glcp	1,5,6-tri-O-acetyl-2,3,4-tri-O-methyl glucitol	16.11	5.12
Galactose	(1→3,6)-Galp	1,3,5,6-tetra-O-acetyl-2,4-tri-O-methyl galactitol	19.61	11.21
Galacturonic Acid	(1→4)-GalpA	1,4,5-tri-O-acetyl-2,3,6-tri-O-methyl galactitol	14.7	13.13
	(1→)-GalpA	1,5-tri-O-acetyl-2,3,4,6-tetra-O-methyl galactitol	10.48	7.65

**Table 3 ijms-24-06247-t003:** ^1^HNMR and ^13^CNMR Chemical Shifts of PPL-1.

Monosaccharide	Molar Ratio	Type of Linkage	H1	H2	H3	H4	H5	H6
			C1	C3	C3	C4	C5	C6
Arabinose	1	(1→2)-linked α-Araf	5.1	3.55	3.85	3.97	4.07	
		(A)	107.41	71.45	74.73	69.57	60.72	
Glucose	13	(1→4)-linked α-Glcp	5.8	3.91	4.64	3.25	3.5	3.39
		(B)	106.84	68.78	78.88	70.14	73.09	70.83
	2	(1→)-linked β-Glcp	4.5	3.25	4.52	3.23	3.44	3.78
		(C)	102.96	74.11	78.36	70.5	73.09	68.07
	1	(1→6)-linked α-Glcp	4.93	4.4	3.27	3.47	3.71	3.77
		(D)	99.86	70.43	77.84	75.92	70.59	72.25
Galactose	2	(1→3,6)-linked α-Galp	5.23	3.46	4.97	4.54	3.27	3.37
		(E)	95.88	76.5	98.57	77.72	67.61	67.61
Galacturonic Acid	3	(1→4)-linked β-GalpA	4.65	3.42	3.7	3.83	4.14	
		(F)	95.75	73.09	72.73	71.42	75.73	170.93
	3	(1→)-linked α-GalpA	5.24	4.6	3.89	4.04	3.76	
		(G)	91.96	79.02	68.64	74.53	70.59	175.41

## Data Availability

Not applicable.

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
