# Peer review of "An Acidic Heteropolysaccharide Isolated from Pueraria lobata and Its Bioactivities"

_ijms, 2023, doi:10.3390/ijms24076247_

Round 1
Reviewer 1 Report
Summary
In this article the authors isolate and characterize a heteropolysaccharide produced by Pueraria lobata. The authors then go on to evaluate its bioactivity and suggest its use as an antioxidant and/or an immunoenhancer.
General Comments
The manuscript does an effective job of characterizing the polysaccharide using NMR and a variety of other analytical techniques. The assessment of the potential bioactivity is a little ‘thin’ and comes across as more of an afterthought. Either additional experiments or a more robust explanation of the data presented is required.
Generally, the language needs a bit of work, nothing too drastic though. There also seems to an issue with word spacing throughout the manuscript.
Line 13
Percentage composition of each monomer should be written in lowest common ratio (and changed throughout where applicable).
Line 65
The total measured carbohydrates + uronic acids is 85%. What accounts from the remaining 15%?
Line 73
There should be a corresponding figure for Mw. This could replace Figure 1B, which is unnecessary.
Line 86
Were there any attempts made to quantify the Selenium? What are typical levels found?
Line 108 - onwards
The reviewer would suggest that pyranose and furanose notations should be in italics.
Line 111-114
The arabinose derivative is missing here. The derivative listed in Table 2 is also incorrectly written (di- instead of tri-).
Line 152-153
Are there any suggestions for the differences in molar ratios between the GC-MS and NMR findings?
Table 3
Please label the monomers/types of linkage A-G for easier reading going forwards.
Figure 3
The water signal is rather high (A, C, D, E). Were alternative solvents or other methods for water suppression explored?
Annotation should be added to spectra A, B & F.
The structure shown in G is incorrect, the reviewer suspects that the branches on the bottom each need to be shifted 1 sugar unit to the left (from the Glc to the Gal).
Line 165
DPPH abbreviation. Spell out first time of usage.
Line 173
Please describe the control group here.
Line 174
What is meant by the ‘nature recover group’? Either explain this or describe in another way.
Line 177
What does Vc stand for?
Figure 5
This figure should be described (and what it shows) in the results section, currently it is only mentioned in the discussion.
Figure 6
The figure legend should be fixed as it is spread across many lines. The descriptions of what B-D show is confusing, this should be described more clearly.
Line 189-191
All of the key characterization features listed in this study (monosaccharides, linkages, Mw) are very different to those previously described by other authors. There are some differences in methodologies used for characterization, as stated in the text, but do you suspect that the differences arise solely from differences in extraction and purification, or could there actually be a number of different polysaccharides present in the roots themselves?
Line 197
Is there a word missing? ‘enzymatic hydrolysis (removes ???)’
Line 206
There is no mention of amylase treatment in the methodology. Please clarify/correct this and provide the type of amylase used.
Line 210
Gal (62%), change to Glc.
Line 221
Could the authors please comment on this huge Mw range.
Line 230
Please reference previous studies.
Line 232
How is the ‘optimal choice’ defined?
Author Response
Dear reviewer,
Thank you very much for your review. We answered the questions.
Best wishes

Reviewer 2 Report
The present study deals with the isolation of an acidic heteropolysaccharide isolated from Pueraria lobata and its bioactivities. The scope of the study is well stated, and statistical analysis of the results is implemented. The results are discussed and compared to the existing literature.
Here are some comments:
- Which was the drying method followed for the roots of Pueraria lobata?
- Where and when were the plants collected?
- Please provide the UPLC chromatogram.
- Please provide details regarding the equipment used, where not provided (e.g spectrophotometer, HPLC etc).
- Pg14, ln 358: Please define Vc
- Pg12, lns 259-264. This part could be transferred under the title “Conclusions”.
- Pg 10: figure 6 description must be corrected.
- Pg 10, ln 193: Please provide the appropriate references.
- Pg 10, ln 193-198. This part could be added to the M&M part, since it is descriptive of the methods used. It does not match to the Discussion part. In addition in line 197 “….enzymatic hydrolysis (removes)…” something is missing.
- Figure 4: the figure legend abbreviations should appear in the same way on the figure (Vc-Vcis) or vice versa.
- English should be rechecked carefully and revised.
Author Response

(The authors gave the same response as above.)

Round 2
Reviewer 2 Report
The authors addressed all the matters arised. The manuscript can be accepted in its current form.